# Absence of association between host genetic mutations in the ORAI1 gene and COVID-19 fatality

**Heba Shawer** [ORCID], **Chew W. Cheng**[ORCID], **Marc A. Bailey** *

The Leeds Institute of Cardiovascular & Metabolic Medicine, School of Medicine, University of Leeds, Leeds, United Kingdom

☯ These authors contributed equally to this work.

\* m.a.bailey@leeds.ac.uk

**Data Availability Statement:** The data underlying the results presented in the study are available from the UK Biobank on application via the UK Biobank online access management system. The

## Abstract

The calcium ion channel ORAI1 has emerged as a promising therapeutic target for the Coronavirus Disease 19 (COVID-19)-associated pneumonia, and a pharmacological inhibitor of ORAI1 has now reached clinical trials for severe COVID-19 pneumonia. Whether ORAI1 itself is associated with an increased risk for severe COVID-19 presentation is still unknown. Here, we employed genetic association analysis to investigate the potential association of host genetic polymorphisms of ORAI1 with the risk of Severe Acute Respiratory Syndrome Coronavirus 2 (SARS-CoV-2) infection and its associated COVID-19 fatality in UK Biobank participants from white British background. The analysis showed no significant association between ORAI1 variants and COVID-19 positivity or fatality, despite the well-established roles of ORAI1 in immune response and inflammation and the success of ORAI1 inhibition in clinical trials. Our results suggest that the host genetic polymorphisms of ORAI1 are unlikely to be implicated in the broad variability in symptoms severity among afflicted patients.

## Background

The emergence of a new, fast-spreading infectious respiratory disease outbreak, at the end of 2019 has presented a global threat to public health with its fast transmission and serious multi-organ complications. The disease is caused by a new virus from the coronavirus species, designated as Severe Acute Respiratory Syndrome Coronavirus 2 (SARS-CoV-2), and the disease is now known as Coronavirus Disease 19 (COVID-19) [1]. The worldwide confirmed COVID-19 cases, as of 23 November 2021, have exceeded 256 million cases and accounts for around 5.1 million deaths [2]. SARS-CoV-2 is a new strain of coronavirus species, similar to SARS-CoV and MERS-CoV that caused more limited outbreaks in 2002 and 2012, respectively. SARS-CoV-2, however, appears to be less deadly compared to SARS-CoV and MERS-CoV but is more transmissible. COVID-19 patients have presented with wide-range of severe symptoms both within and beyond the respiratory system, ranging from no or mild symptoms, to severe acute pneumonia, respiratory failure, and could be even life threatening. COVID-19 was also

UK Biobank data is available to all bona fide researchers by applying through the UK Biobank Access Management System (www.ukbiobank.ac.uk/register-apply). Details on how to complete an application to access the UK Biobank research resource are available here: https://www.ukbiobank.ac.uk/media/px5gbq4q/access_019-access-management-system-user-guide-v4-1.pdf. The authors did not have any special access privileges.

**Funding:** This work was supported by British Heart Foundation fellowships to MAB and HS (FS/18/12/33270, FS/17/66/33480) and Leeds Cardiovascular Endowment support to CWC.

**Competing interests:** The authors have declared that no competing interests exist.

reported to be associated with non-respiratory complications, including myocardial injury, multisystem inflammatory syndrome, and haematological and thrombotic complications [3–5]. To this end, there has always been a question regarding the source of this large range of variation in symptoms severity. Failure to answer this question has hindered the true identification of individuals at high risk of serious illness, as well as those who are likely to be asymptomatic carriers and unknowingly spreading the virus. To date, the pathophysiology underlying this varied manifestation of the disease is still elusive. Age [6, 7], gender [8], obesity [9], and comorbid conditions [7, 10, 11] appeared to be risk factors associated with adverse COVID-19 outcomes. Nonetheless, severe illness was worryingly observed in young, otherwise healthy, individuals [12, 13]. Discrepancies in disease burden and the risk of infection was also observed among the different ethnic groups [14–16]. A possible explanation of the substantial variation in susceptibility to SARS-CoV-2 virus and the wide unpredictability in COVID-19 manifestation is the variability in the host genetic background. In fact, emerging genome-wide association [17, 18] and candidate gene association [19] studies have revealed an association between a host genetic risk factor and the severity of COVID-19 illness.

Here, we investigate the host genetic polymorphisms within the ORAI1 gene that could be implicated in the genetic risk for SARS-CoV-2 infection and for the severity of COVID-19 symptoms. The ORAI1 gene located at the long arm of Chr12 at 12q24.31 is a gene encoding for the calcium ion channel ORAI1, which is the *de facto* pore forming ion channel mediating store operated calcium entry (SOCE). ORAI1 is an interesting candidate gene in the context of COVID-19 as it is known to play a key role in the immune response, inflammation, platelet activation and thrombus formation. As innate immune cell over-activation, an exacerbated inflammatory response and thrombosis were linked with adverse COVID-19 outcome, ORAI1 could potentially play a role in this over-exuberant immune response, thrombotic state and adverse outcome in cases with serious symptoms and therefore could be a promising therapeutic target for COVID-19. This is not a new suggestion. The pharmacological ORAI1 inhibitor Auxora, produced by CalciMedica, has reached clinical trials for severe COVID-19 pneumonia [20]. In a 2:1 randomized, open label trial of patients with severe COVID-19 pneumonia, the intravenous administration of Auxora, showed promising safety profile and favourable outcomes in patients with severe COVID-19 pneumonia. Its therapeutic benefits are currently being further investigated in a randomized, placebo-controlled, double-blind study. Mutations within ORAI1 originally manifested as severe combined immune deficiency (SCID) [21]. It is however still unknown if ORAI1 small nucleotide polymorphisms (SNPs) could be implicated in an increased risk of SARS-CoV-2 infection or COVID-19-associated fatality. Our aim is to study the implications of host genetic polymorphisms within the ORAI1 gene in the genetic risk for SARS-CoV-2 infection and for the severity of COVID-19 symptoms. This information could help identify individuals at high risk of adverse COVID-19 outcomes and reveal potential therapeutic target. Therefore, this candidate gene association study investigates the potential associations of polymorphisms within the ORAI1 gene and the susceptibility to SARS-CoV-2 infection as well as COVID-19 fatality in UK Biobank dataset.

## Methods

### Participants

UK Biobank dataset of participants tested for COVID-19 (January 2021 release) was utilised under UK Biobank accession number 60315. UK Biobank received ethical approval from the North West Multi-Centre Research Ethics Committee and was conducted in accordance with the principles of the Declaration of Helsinki. No separate ethical approval was required. This dataset comprises 47,988 participants tested for COVID-19. As the majority of participants are

**Table 1. Basic characteristics of the study participants.**

| Characteristics | All | COVID-19 Positive | COVID-19 Negative | COVID-19 Fatalities | Non-fatal COVID-19 |
|---|---|---|---|---|---|
| n* | 41146 | 7462 | 33684 | 332 | 7130 |
| Mean age, years ± SD** | 68.8 ± 8 | 68.8 ± 8.1 | 68.7 ± 8 | 69.1 ± 8.4 | 68.8 ± 8.1 |
| **Gender** | | | | | |
| Female, n (%) | 22076 (53.65%) | 3978 (53.3%) | 18098 (53.7%) | 181 (54.52%) | 3797 (53.25%) |
| Male, n (%) | 19070 (46.35%) | 3484 (46.7%) | 15586 (46.27%) | 151 (45.48%) | 3333 (46.75%) |
| **Co-morbidities** | | | | | |
| Stroke, n (%) | 770 (1.87%) | 149 (2.0%) | 621 (1.84%) | 11 (3.31%) | 138 (1.94%) |
| Hypertension, n (%) | 12788 (31.08%) | 2018 (27.04%) | 10770 (31.97) | 163 (49.1%) | 1855 (26.02%) |
| Diabetes, n (%) | 2520 (6.12%) | 417 (5.59%) | 2103 (6.24%) | 52 (15.66%) | 365 (5.12%) |
| Asthma, n (%) | 5562 (13.52%) | 1020 (%) | 4542 (13.48%) | 34 (10.24%) | 986 (13.83%) |
| Heart/cardiac problem, n (%) | 210 (0.51%) | 38 (13.67%) | 172 (0.51%) | 4 (1.20%) | 34 (0.48%) |
| Myocardial infarction, n (%) | 1364 (3.32%) | 229 (3.07%) | 1135 (3.37%) | 26 (7.83%) | 203 (2.85%) |
| Chronic Obstructive pulmonary disease, n (%) | 213 (0.52%) | 39 (0.52%) | 174 (0.52%) | 3 (0.90%) | 36 (0.50%) |

*n, number of individuals,

**SD, standard deviation.

from British decent, the analysis was restricted to this group. After applying the filtering parameters, among the 41,146 individuals from British ancestor, 7,462 (18.1%) were reported to have been diagnosed with COVID-19, and 33,684 (81.9%) were negative controls. The death records of COVID-19 positive cases showed 332 (4.4%) COVID-19 deaths and 7,130 (95.6%) COVID-19 survivals. Participants' characteristics are summarised in Table 1.

## Genetic association analysis

UK Biobank imputed genomic data of Chr12 were obtained and variants with imputation quality (Info score) < 0.4 were filtered out. Details about UK Biobank genome-wide genotyping and the genotypes imputation were previously described in [22, 23]. Quality measures were applied to exclude participants with more than 10% missing data, exclude variants with Hardy-Weinberg equilibrium (HWE) less than $1x10^{-6}$ and exclude variants with minor allele frequency (MAF) less than 5% for common variants or less than 1% for less frequent variants, using PLINK version 2.0 software [24]. After applying the filtering parameters, 7,462 COVID-19 positive cases and 33,684 control cases were retained and eligible for downstream analyses. Filtered variants spanning Chr12 (312,261 variants) were then examined for genetic association with COVID-19 positivity and fatality (Fig 1). The Manhattan pots were generated using the qqman R package and the regional association plots were generated using LocusZoom (http://locuszoom.sph.umich.edu) [25].

## Statistics

After adjusting for effects of sex, age, and the first ten principal components that conveys variations in population structure, the associations of variants within ORAI1 (chr12:122,064,455– 122,080,583, GRCh37/hg19) with COVID-19 positivity (7,462 cases and 33,684 controls) and fatality (332 deaths and 7,130 controls) were examined using logistic regression, conducted using PLINK version 2.0 [24]. Bonferroni correction was applied on the P-value to adjust for the number of variants tested. Bonferroni corrected P-value of 0.05 was used as a threshold to indicate statistical significance.

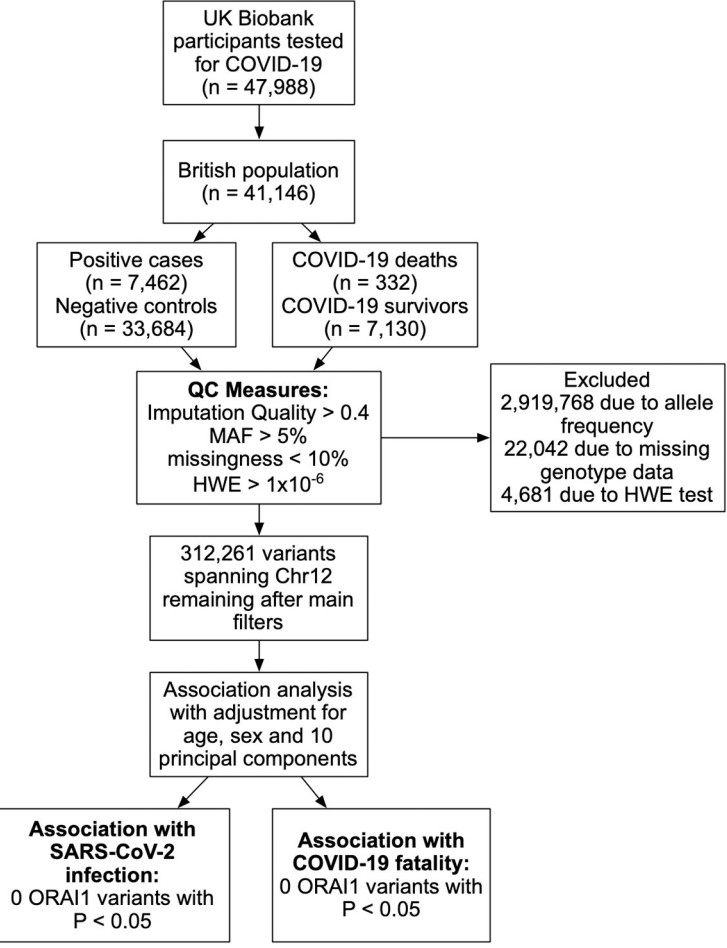

**Fig 1. Flowchart of the analysis pipeline.** ORAI1 variants associations with SARS-CoV-2 infection and COVID-19 fatality were examined in British UK Biobank participants. There was no significant association observed between ORAI1 variants and COVID-19 positivity or fatality.

## Results

### Participants characteristics

Data collected from 41,146 participants of British ancestry, tested for COVID-19, were obtained from the UK Biobank (Fig 1). Table 1 shows the basic characteristics of the study participants. The study participants were 46% males (average age 69) and 54% females (average age 68.6), with overall mean age of 68.8 years (age range from 50 to 84 years). In total, 332 COVID-19 deaths were reported, with mean age of 69.1 years, of which 45.5% are males with mean age of 69.8, and 54.5% are females with mean age of 68.7. On the other hand, non-fatal COVID-19 cases were 47% males (average age 69 years), and 53% females (average age 68.7 years) and all non-fatal cases had an average age of 68.8 years. We did not observe substantial differences between the mean age of fatal (69.1 ± 8.4 years) and non-fatal (68.8 ± 8.1 years) COVID-19 cases, which could be attributed to the confined age range of the participants between 50 and 84 years. Therefore, the younger age group with a lower risk of serious COVID-19 illness was not captured in our analysis. As expected, the incidence of comorbid conditions associated with severe COVID-19 symptoms was more common among fatal compared to non-fatal COVID-19, with 49.1% of fatal, 26.02% of non-fatal COVID-19 cases had

concomitant hypertension, 3.31% of fatal and 1.94% of non-fatal cases were diagnosed with previous stroke, 15.66% of fatal while 5.12% of non-fatal COVID-19 cases were diagnosed with diabetes, 10.24% of fatal and 13.83% of non-fatal COVID-19 cases were diagnosed with asthma, and 7.83% of fatal, while 2.85% of non-fatal COVID-19 cases were diagnosed with myocardial infarction (Table 1).

## SARS-CoV-2 infection and COVID-19 fatality

The genetic variants within the ORAI1 gene were examined for association with COVID-19 positivity. There was no statistically significant association between SNPs within the ORAI1 locus and SARS-CoV-2 infection (Fig 2, Table 2). We then examined the association between ORAI1 variants and COVID-19 fatality. The analysis was performed in 332 fatal and 7130 non-fatal COVID-19 cases. There was also no significant association between ORAI1 variants

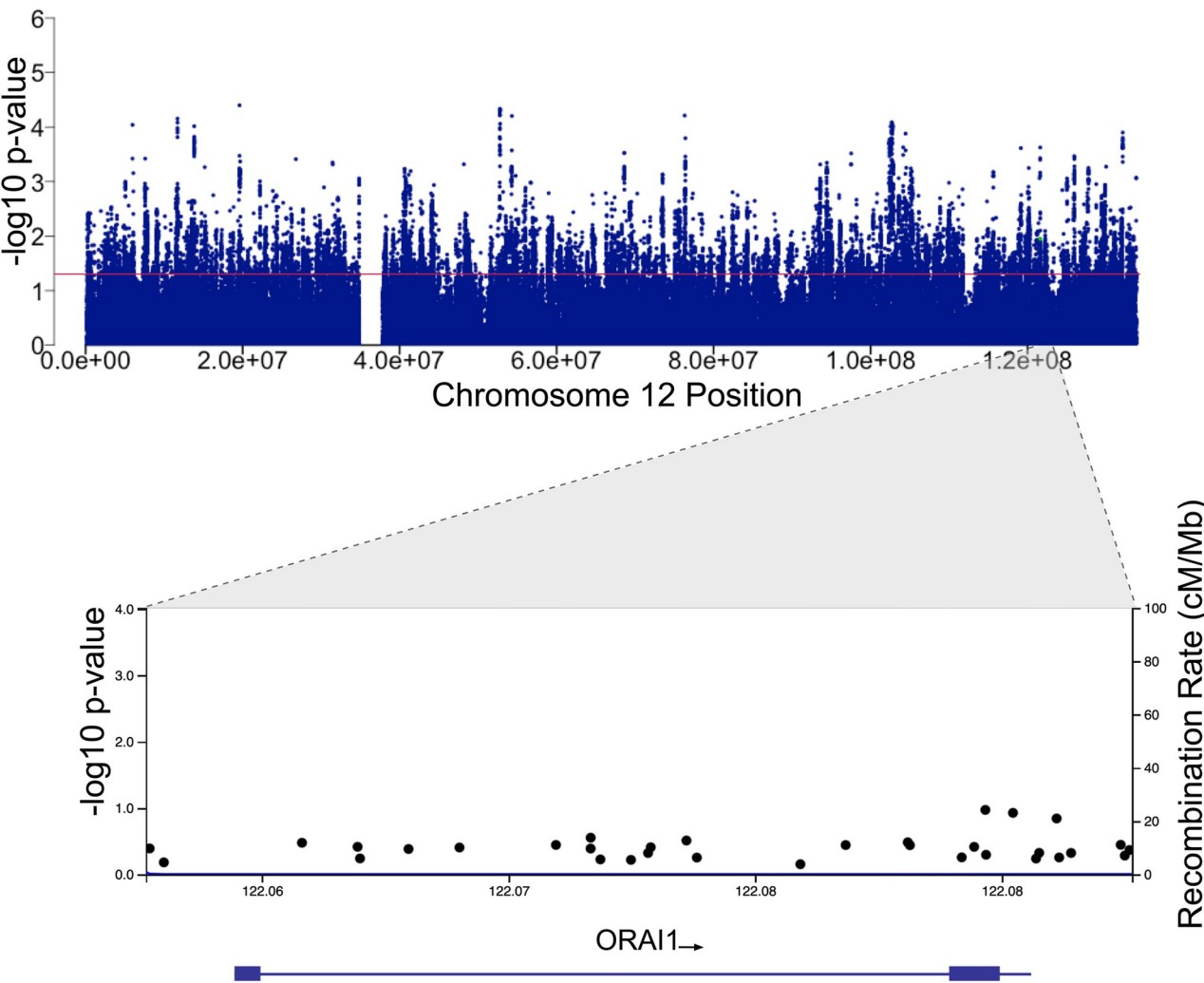

**Fig 2. Candidate gene association analysis of the ORAI1 locus in SARS-CoV-2 positive cases.** Manhattan plot showing the association of SNPs located within Chr12 with SARS-CoV-2 positivity. The red line indicates a P value of 0.05. Regional association plot at 12q24 showing the SNPs within the region comprising the ORAI1 locus. SNPs are plotted according to their chromosomal position on build GRCh37/hg19 displayed on the x-axis and the –$\log_{10}$ scale of the P-value of their association with SARS-CoV-2 positivity, plotted on the left y-axis. The right y-axis displays the recombination rate estimated from the 1000 Genomes European data.

**Table 2. The association of common ORAI1 variants with MAF more than 5% with SARS-CoV-2 positivity.**

| SNP ID | REF | ALT | OR | LOG(OR)_SE | L95 | U95 | Z_STAT | P |
|--------|-----|-----|------|-----------|------|------|--------|------|
| rs7963749 | C | G | 1.04 | 0.04 | 0.96 | 1.12 | 0.97 | 0.33 |
| rs6486786 | T | C | 1.02 | 0.02 | 0.98 | 1.05 | 0.87 | 0.38 |
| rs6486787 | G | C | 0.99 | 0.02 | 0.94 | 1.03 | -0.56 | 0.58 |
| rs6486789 | T | C | 1.02 | 0.02 | 0.98 | 1.05 | 0.82 | 0.42 |
| rs7135617 | T | G | 1.02 | 0.02 | 0.98 | 1.05 | 0.85 | 0.39 |
| rs7484839 | C | T | 0.98 | 0.03 | 0.93 | 1.03 | -0.91 | 0.36 |
| rs7968061 | T | C | 0.98 | 0.03 | 0.93 | 1.03 | -0.83 | 0.41 |
| rs3892486 | G | A | 0.97 | 0.03 | 0.92 | 1.02 | -1.08 | 0.28 |
| rs7956644 | G | A | 0.99 | 0.02 | 0.94 | 1.03 | -0.53 | 0.60 |
| rs6486790 | A | G | 0.99 | 0.02 | 0.94 | 1.03 | -0.52 | 0.61 |
| rs7398511 | C | A | 0.98 | 0.02 | 0.94 | 1.03 | -0.71 | 0.48 |
| rs10522094 | C | CAGGG | 1.02 | 0.02 | 0.98 | 1.05 | 0.86 | 0.39 |
| rs139720017 | G | GT | 1.03 | 0.03 | 0.97 | 1.10 | 1.02 | 0.31 |
| rs12300327 | A | G | 0.99 | 0.02 | 0.94 | 1.03 | -0.58 | 0.56 |
| rs6486795 | T | C | 0.99 | 0.02 | 0.95 | 1.04 | -0.38 | 0.70 |
| rs11043296 | C | T | 0.98 | 0.03 | 0.93 | 1.03 | -0.91 | 0.36 |
| rs11043305 | G | A | 1.03 | 0.03 | 0.97 | 1.09 | 0.98 | 0.33 |
| rs11043306 | T | C | 0.98 | 0.03 | 0.93 | 1.03 | -0.91 | 0.36 |
| rs3741595 | C | T | 0.99 | 0.02 | 0.94 | 1.03 | -0.59 | 0.56 |
| rs3825175 | T | C | 1.02 | 0.02 | 0.98 | 1.05 | 0.87 | 0.39 |
| rs712853 | A | G | 1.03 | 0.02 | 0.99 | 1.08 | 1.61 | 0.11 |
| rs35558190 | A | AT | 0.98 | 0.02 | 0.94 | 1.03 | -0.66 | 0.51 |
| rs1983268 | G | A | 1.03 | 0.02 | 0.99 | 1.08 | 1.56 | 0.12 |

REF, reference allele; ALT, alternative allele; OR, Odds Ratio; L95, lower 95% confidence interval; U95, upper 95% confidence interval.

and COVID-19 fatality (Fig 3, Table 3), despite its role in immune cell function and the promising outcome of ORAI1 inhibition in the randomized, controlled, open-label study conducted in patients with severe or critical COVID-19 pneumonia [20]. We did not observe significant association between common variants with MAF > 5% or less frequent variants with MAF > 1% in the ORAI1 gene with COVID-19 positivity or fatality in the examined participants.

## Discussion

Defining the host genetic polymorphisms that contributes to COVID-19 severity could help unravel causes of the observed inter-individual variability of COVID-19 symptoms and death. This could help aid genetic counselling for our afflicted patients and improve our preventative strategies. We investigated the potential association between polymorphisms within the candidate gene, ORAI1, with the susceptibility to SARS-CoV-2 infection and the disease-associated fatality in UK Biobank white British participants. We observed no significant association between ORAI1 variants and COVID-19 positivity or fatality in our examined population.

The human Chr12 harbours over 1,400 protein coding genes, and a number of them are implicated in immune response and inflammation, which are essential processes to fight any infection, including SARS-CoV-2 infection [26]. Among the genes located on Chr12 is the ORAI1 gene that encodes for the SOCE ORAI1 channel. The main reported clinical manifestation of the ORAI1 loss-of-function mutations in patients is immunodeficiency resulting from

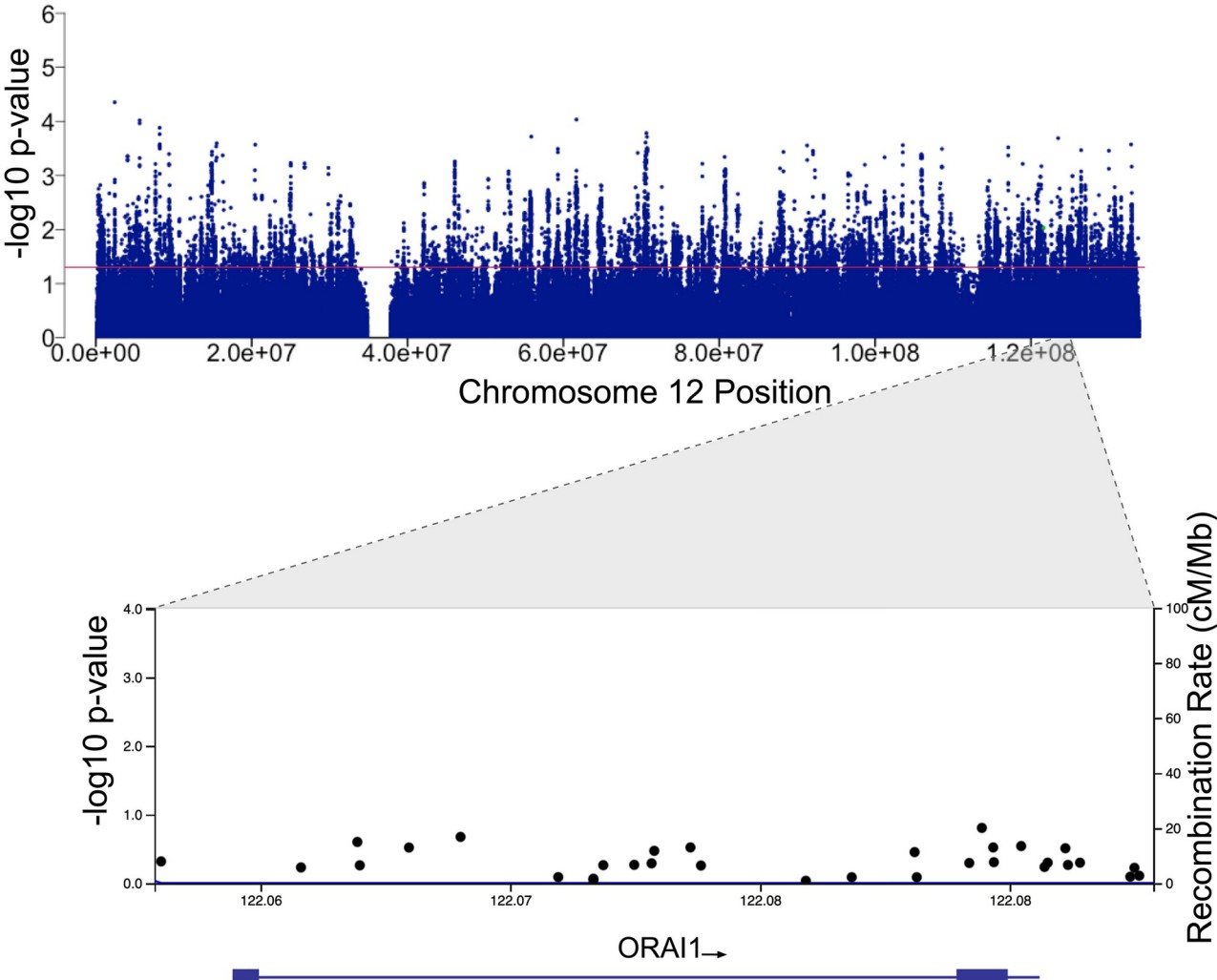

**Fig 3. Association analysis of variants within the ORAI1 locus with COVID-19 fatality.** Manhattan plot showing the association of SNPs located within Chr12 with COVID-19 fatality. The red line indicates a P-value of 0.05. Regional association plot at 12q24 showing the SNPs within the region comprising the ORAI1 locus. SNPs are plotted according to their chromosomal position on build GRCh37/hg19 displayed on the x-axis and the–$\log_{10}$ scale of the P-value of their association with COVID-19 fatality, plotted on the left y-axis. The right y-axis displays the recombination rate estimated from the 1000 Genomes European data.

impaired T cell activation and cytokine production [21, 27, 28], which highlights the fundamental role of ORAI1 in immune response against pathogens. ORAI1 channel activity was also shown to be a major player in inflammation and cytokine production [29]. This implication of ORAI1 channels in inflammation was previously observed in pulmonary endothelial cell injury, increased pulmonary permeability and alveolar-vascular barrier dysfunction [30–32]. The linkage between ORAI1, the immune, and inflammatory responses, as well as the fact that strong evidence implicates a hyper-inflammatory state in the adverse COVID-19 outcome [33, 34], foreshadow a potential role of ORAI1 in the pro-inflammatory cytokine storm reported in severe COVID-19 illness. This is further supported by the positive outcomes observed in clinical trial of the ORAI1 inhibitor Auxora in patients with severe COVID-19 pneumonia [20].

**Table 3. Non-significant association of common ORAI1 variants with MAF more than 5% with COVID-19 fatality.**

| SNP ID | REF | ALT | OR | LOG(OR)_SE | L95 | U95 | Z_STAT | P |
|---|---|---|---|---|---|---|---|---|
| rs7963749 | C | G | 1.09 | 0.17 | 0.79 | 1.52 | 0.54 | 0.59 |
| rs6486786 | T | C | 0.91 | 0.08 | 0.78 | 1.07 | -1.15 | 0.25 |
| rs6486787 | G | C | 0.94 | 0.10 | 0.76 | 1.15 | -0.60 | 0.55 |
| rs6486789 | T | C | 0.92 | 0.08 | 0.78 | 1.08 | -1.03 | 0.30 |
| rs7135617 | T | G | 0.90 | 0.08 | 0.77 | 1.06 | -1.25 | 0.21 |
| rs7484839 | C | T | 0.97 | 0.12 | 0.77 | 1.22 | -0.23 | 0.81 |
| rs7968061 | T | C | 0.98 | 0.11 | 0.79 | 1.22 | -0.18 | 0.86 |
| rs3892486 | G | A | 0.99 | 0.11 | 0.79 | 1.23 | -0.13 | 0.90 |
| rs7956644 | G | A | 0.94 | 0.11 | 0.76 | 1.15 | -0.60 | 0.55 |
| rs6486790 | A | G | 0.94 | 0.11 | 0.76 | 1.15 | -0.61 | 0.54 |
| rs7398511 | C | A | 0.93 | 0.11 | 0.75 | 1.15 | -0.65 | 0.51 |
| rs10522094 | C | CAGGG | 0.92 | 0.08 | 0.79 | 1.09 | -0.96 | 0.34 |
| rs139720017 | G | GT | 1.14 | 0.13 | 0.89 | 1.47 | 1.03 | 0.30 |
| rs12300327 | A | G | 0.94 | 0.10 | 0.76 | 1.15 | -0.59 | 0.55 |
| rs6486795 | T | C | 0.99 | 0.10 | 0.81 | 1.20 | -0.10 | 0.92 |
| rs11043296 | C | T | 0.97 | 0.12 | 0.77 | 1.22 | -0.23 | 0.82 |
| rs11043305 | G | A | 1.13 | 0.13 | 0.88 | 1.45 | 0.93 | 0.35 |
| rs11043306 | T | C | 0.97 | 0.12 | 0.77 | 1.22 | -0.23 | 0.82 |
| rs3741595 | C | T | 0.93 | 0.11 | 0.75 | 1.16 | -0.66 | 0.51 |
| rs3825175 | T | C | 0.89 | 0.08 | 0.76 | 1.05 | -1.42 | 0.16 |
| rs712853 | A | G | 0.91 | 0.09 | 0.76 | 1.09 | -1.03 | 0.30 |
| rs35558190 | A | AT | 0.93 | 0.11 | 0.75 | 1.15 | -0.68 | 0.49 |
| rs1983268 | G | A | 0.90 | 0.10 | 0.75 | 1.09 | -1.06 | 0.29 |

REF, reference allele; ALT, alternative allele; OR, Odds Ratio; L95, lower 95% confidence interval; U95, upper 95% confidence interval.

The well-established roles of ORAI1 in immune response and inflammation rendered it as a promising candidate gene that could be implicated in an altered susceptibility to COVID-19 positivity and disease severity. This important role in immune cell function and inflammation motivated us to examine the potential association between ORAI1 polymorphisms and the severity of COVID-19 illness. Nonetheless, there was no significant association observed between ORAI1 variants and COVID-19 positivity or fatality in British UK Biobank participants, which could be attributed to the small number of cases in the study. The observed absence of genetic association between ORAI1 variants and COVID-19 fatality does not eliminate the possible implication of ORAI1 in disease pathogenesis through altered gene expression levels or channel activity and therefore it does not eliminate the potential usefulness of targeting ORAI1 as a therapeutic target for COVID-19. Our analysis was constrained to participants from white British (Caucasian) background, due to the small number of cases from other ethnic groups in this dataset. Therefore, an impact of ORAI1 variants on COVID-19 fatality in different ethnic groups cannot be ruled out. Further studies are needed to investigate whether ORAI1 polymorphisms influence the susceptibility to COVID-19 positivity or fatality in different ethnic populations. Additionally, the age range of the participants was confined to age 50–84 years and did not include the younger age group with the lower risk of adverse COVID-19 illness. Because of these geographical, ethnic and age limitations, our finding may not reflect the wider more diverse population. Our analysis was also limited by the lack of information about the vaccination status of the participants and about the prevalence of the SARS-CoV-2 variants of concern in the examined population. Further research is still needed

to understand the role of ORAI1 in severe COVID-19 illness and to investigate the potential genetic association of ORAI1 variants and adverse COVID-19 outcome in different populations.

## Acknowledgments

This work was conducted using UK Biobank datasets. We thank the UK Biobank participants and staff. This work was undertaken on ARC3, part of the High-Performance Computing (HPC) facilities at the University of Leeds, UK.

## Author Contributions

**Conceptualization:** Heba Shawer, Chew W. Cheng, Marc A. Bailey.

**Formal analysis:** Heba Shawer, Chew W. Cheng.

**Funding acquisition:** Marc A. Bailey.

**Investigation:** Heba Shawer, Chew W. Cheng, Marc A. Bailey.

**Methodology:** Heba Shawer, Chew W. Cheng.

**Project administration:** Marc A. Bailey.

**Supervision:** Marc A. Bailey.

**Writing – original draft:** Heba Shawer, Chew W. Cheng, Marc A. Bailey.

**Writing – review & editing:** Heba Shawer, Chew W. Cheng, Marc A. Bailey.

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
