## [Decision Letter · Decision Letter 0]

3 Nov 2021

PONE-D-21-24713Absence of association between host genetic mutations in the ORAI1 gene and COVID-19 fatalityPLOS ONE

Dear Dr. Bailey,

Thank you for submitting your manuscript to PLOS ONE. After careful consideration, we feel that it has merit but does not fully meet PLOS ONE’s publication criteria as it currently stands. Therefore, we invite you to submit a revised version of the manuscript that addresses the points raised during the review process.

We look forward to receiving your revised manuscript.

Kind regards,

Alessandro Marchioni

Academic Editor

PLOS ONE

“This work was supported by British Heart Foundation fellowships to MAB and HS (FS/18/12/33270, FS/17/66/33480) and Leeds Cardiovascular Endowment support to CWC. This work was conducted using UK Biobank datasets. We thank the UK Biobank participants and staff. This work was undertaken on ARC3, part of the High-Performance Computing (HPC) facilities at the University of Leeds, UK.”

“This work was supported by British Heart Foundation fellowships to MAB and HS (FS/18/12/33270, FS/17/66/33480) and Leeds Cardiovascular Endowment support to CWC.”

Reviewers' comments:

Reviewer's Responses to Questions

**Comments to the Author**

1. Is the manuscript technically sound, and do the data support the conclusions?

Reviewer #1: Yes

Reviewer #2: Yes

2. Has the statistical analysis been performed appropriately and rigorously? 

Reviewer #1: Yes

Reviewer #2: I Don't Know

3. Have the authors made all data underlying the findings in their manuscript fully available?

Reviewer #1: Yes

Reviewer #2: No

4. Is the manuscript presented in an intelligible fashion and written in standard English?

Reviewer #1: Yes

Reviewer #2: Yes

5. Review Comments to the Author

Reviewer #1: Study based on a biobank dataset to clarify the role of genetic variants within the ORAI1 gene with positivity and mortality for COVID-19. The study design is not clear, the authors do not accurately describe what the main objective is; these aspects should be better defined.

The results demonstrate that there is no significant association between ORAI1 variants and COVID-19 positivity or fatality in the study participants, despite the well-established roles of ORAI1 in immune response and inflammation and the successful inhibition of ORAI1 in clinical studies. Authors should explain the reason for this discrepancy in more detail than they have already done.

Reviewer #2: I am grateful for the chance to review this work by Heba and coworker. The authors retrospectively employed genetic association analysis to investigate the potential association of host genetic 18 polymorphisms of ORAI1 with the risk of developing COVID-19 with COVID-19 fatality in 41146 with British individuals. They found that there was no significant association observed between ORAI1 199 variants and COVID-19 positivity or fatality.

The research question is extremely appealing being the susceptibility to SARS-CoV-2 virus quite unpredictable and given the promising results of the ORAI1 inhibitor in patients with COVID-19. Since study aim is extremely relevant, methodological procedures are correct and statistical analysis is well described and convincing. Manuscript sections are well organized and clear.

I just have a few issues to address:

1) Introduction: in my opinion it should be shortened and focused on the research question.

2) Methods: the frequency of ORAI1 SNpolymorphism should be detailed and considered to estimate a sample large enough to test the research question.

3) Methods: did the authors collected data regarding SARS-CoV-2 major variants? If so, wouldn’t it be interesting to test whether an association with ORAI1 SNP could be find?

4) Methods: the authors should report the percentage of vaccinated people.

4) Results: it is surprising that COVID-19 fatal and non-fatal case have similar average age, gender distribution and hypertension rate. These data are in contrast with the great majority of results reported in literature and should be discussed in more details.

5) Results: Data regarding the prevalence of hearth problem in the COVID-19 fatality group should be checked (just 1 over 332?).

6. PLOS authors have the option to publish the peer review history of their article (what does this mean?). If published, this will include your full peer review and any attached files.

Reviewer #1: No

Reviewer #2: No

---

## [Author Response · Author response to Decision Letter 0]

3 Dec 2021

Dear Dr Marchioni, 

Thank you for allowing us to respond and provide a revised manuscript to PLOS ONE. We are grateful to Reviewer #1 and Reviewer #2 for the constructive comments, which after addressing, the quality of the manuscript has significantly improved. We have responded to each comment and submitted a tracked version of the manuscript.

Thank you once again for allowing us to resubmit this revised and improved manuscript. 

Yours sincerely, 

Heba Shawer on behalf of all authors

Editor

Response: The manuscript style requirements were reviewed in the manuscript.

2. In your Data Availability statement, you have not specified where the minimal data set underlying the results described in your manuscript can be found. PLOS defines a study's minimal data set as the underlying data used to reach the conclusions drawn in the manuscript and any additional data required to replicate the reported study findings in their entirety. All PLOS journals require that the minimal data set be made fully available. For more information about our data policy, please see http://journals.plos.org/plosone/s/data-availability

Response: Thanks for your comments. The following Data Availability Statement describes where the data set underlying the analysis can be found.

“The data underlying the results presented in the study are available from the UK Biobank on application via the UK Biobank online access management system (http://www.ukbiobank.ac.uk).”

3. Thank you for stating the following in the Acknowledgments Section of your manuscript: “This work was supported by British Heart Foundation fellowships to MAB and HS (FS/18/12/33270, FS/17/66/33480) and Leeds Cardiovascular Endowment support to CWC. This work was conducted using UK Biobank datasets. We thank the UK Biobank participants and staff. This work was undertaken on ARC3, part of the High-Performance Computing (HPC) facilities at the University of Leeds, UK.”

Response: We appreciate the comments. The funding statement is now removed from the Acknowledgments Section to be only stated in the Funding Section.

Reviewer #1

1. Study based on a biobank dataset to clarify the role of genetic variants within the ORAI1 gene with positivity and mortality for COVID-19. The study design is not clear, the authors do not accurately describe what the main objective is; these aspects should be better defined. 

Response: We have more clearly defined the rationale for the study, the design and our objective on page 3, line 73-79, in the untracked version of the revised manuscript.

2. The results demonstrate that there is no significant association between ORAI1 variants and COVID-19 positivity or fatality in the study participants, despite the well-established roles of ORAI1 in immune response and inflammation and the successful inhibition of ORAI1 in clinical studies. Authors should explain the reason for this discrepancy in more detail than they have already done.

Response: We added the following clarification in the discussion to explain the potential reasons for not finding significant evidence that ORAI1 variants influence the susceptibility to COVID-19 positivity or fatality.

“This absence of genetic association between ORAI1 variants and COVID-19 fatality doesn’t eliminate the possible implication of ORAI1 in disease pathogenesis through altered gene expression levels or channel activity and therefore it does not eliminate the potential usefulness of targeting ORAI1 as a therapeutic target for COVID-19. Our analysis was constrained to participants from White British (Caucasian) background, due to the small number of cases from other ethnic groups in this data set, and therefore possible implication of ORAI1 variants in COVID-19 fatality in different ethnic groups couldn’t be ruled out. Further studies are needed to investigate whether ORAI1 polymorphisms influence the susceptibility to COVID19 positivity or fatality in different ethnic populations. Additionally, the age range of participants was confined to age 50–84 years and didn’t include the younger age group with the lower risk of adverse COVID-19 illness. Because of these geographical, ethnic and age limitations, our finding may not reflect the wider more diverse population.” Page 10-11, Line 199-211 in the untracked version of the revised manuscript.

Reviewer #2

1. Introduction: in my opinion it should be shortened and focused on the research question.

Response: We have now shorted the introduction for brevity. 

2. Methods: the frequency of ORAI1 SNP polymorphism should be detailed and considered to estimate a sample large enough to test the research question.

Response: To evaluate the effect of the variants frequencies in this locus, we included in our analysis the ORAI1 common variants with minor allele frequency more than 5%, as well as the less frequent variants with minor allele frequency more than 1%. We found no significant evidence that either the common (MAF > 5%) or rare (MAF > 1%) variants influence the susceptibility to COVID-19 positivity or fatality. We have added the following remarks to the revised manuscript to highlight this point. 

“We did not observe significant association between common variants with MAF > 5% or less frequent variants with MAF > 1% in the ORAI1 gene with COVID-19 positivity or fatality in the examined population.” Page 7, Lines 145-147 in the untracked version of the revised manuscript.

3. Methods: did the authors collected data regarding SARS-CoV-2 major variants? If so, wouldn’t it be interesting to test whether an association with ORAI1 SNP could be find?

Response: We appreciate reviewer’s suggestion. Although it is interesting to link ORAI1 SNPs with the major SARS-CoV-2 variants, these data are currently not available in the UK Biobank database. Therefore, studying the association between ORAI1 SNPs in this cohort and SARS-CoV-2 variants is currently not possible. We added the following remarks in the discussion to highlight this limitation in the revised manuscript.

“Our analysis was also limited by the lack of information about the vaccination status of the participants and about the prevalence of the SARS-CoV-2 variants of concern in the examined population.” Page 11, Lines 211-213 in the untracked version of the revised manuscript.

4. Methods: the authors should report the percentage of vaccinated people.

Response: This genetic data is approved and provided by Public Health England through UK Biobank. Unfortunately, the vaccination data is yet to be made available to the scientific community. We appreciate this could be a limitation in the study. This limitation is now highlighted in the manuscript. Page 11, Lines 211-213 in the untracked version of the revised manuscript.

5. Results: it is surprising that COVID-19 fatal and non-fatal case have similar average age, gender distribution and hypertension rate. These data are in contrast with the great majority of results reported in literature and should be discussed in more details.

Response: it is true that the age, gender and number of cases with hypertension were comparable in the COVID-19 fatal and non-fatal groups. This could be attributed to the fact that the UK Biobank dataset contains the genetic data for individuals aged between 50 to 84 and only a subset of COVID-19 data was released at the time of analysis. To clarify this and recognise the limitations, we added the following remarks to the revised manuscript.

“We did not observe substantial differences between the mean age of fatal (69.1 ± 8.4 years) and non-fatal (68.8 ± 8.1 years) COVID-19 cases, which could be attributed to the confined age range of the participants between 50 and 84 years. Therefore, the younger age group with a lower risk of serious COVID-19 illness was not captured in our analysis.”. Page 5, Lines 120-124 in the untracked version of the revised manuscript..

“The age range of participants was confined to age 50–84 years and didn’t include the younger age group with the low risk of adverse COVID-19 illness. Because of these geographical, ethnic and age limitations of the examined dataset, our finding may not reflect the wider more diverse population.” Page 11, Lines 208-211 in the untracked version of the revised manuscript.

6. Results: Data regarding the prevalence of hearth problem in the COVID-19 fatality group should be checked (just 1 over 332?).

Response: Thanks for your comment. We have noticed a calculation error that attributes to this low number of cases with heart problem. We took the opportunity to correct this error and further examine the prevalence of additional more specific and relevant co-morbidities (Myocardial infarction and Chronic Obstructive pulmonary disease) in the examined population. Table 1 with the basic characteristics of the study participants is now updated. Page 6-7 in the untracked version of the revised manuscript.

Table 1. Basic characteristics of the study participants

characteristic All COVID-19 Positive COVID-19 Negative COVID-19 Fatalities Non-fatal COVID-19 

N* 41146 7462 33684 332 7130

Mean age, years ± SD** 68.8 ± 8 68.8 ± 8.1 68.7 ± 8 69.1 ± 8.4 68.8 ± 8.1

Gender

Female, n (%) 22076 (53.65%) 3978 (53.3%) 18098(53.7%) 181 (54.52%) 3797 (53.25%)

Male, n (%) 19070 (46.35%) 3484 (46.7%) 15586 (46.27%) 151 (45.48%) 3333 (46.75%)

Co-morbidities 

Stroke, n (%) 770 (1.87%) 149 (2.0%) 621 (1.84%) 11 (3.31%) 138 (1.94%)

Hypertension, n (%) 12788 (31.08%) 2018 (27.04%) 10770 (31.97) 163 (49.1%) 1855 (26.02%)

Diabetes, n (%) 2520 (6.12%) 417 (5.59%) 2103 (6.24%) 52 (15.66%) 365 (5.12%)

Asthma, n (%) 5562 (13.52%) 1020 (%) 4542 (13.48%) 34 (10.24%) 986 (13.83%)

Heart/cardiac problem, n (%) 210 (0.51%) 38 (13.67%) 172 (0.51%) 4 (1.20%) 34 (0.48%)

Myocardial infarction, n (%) 1364 (3.32%) 229 (3.07%) 1135 (3.37%) 26 (7.83%) 203 (2.85%)

Chronic Obstructive pulmonary disease, n (%) 213 (0.52%) 39 (0.52%) 174 (0.52%) 3 (0.90%) 36 (0.50%)

*N, number of individuals, **SD, standard deviation

---

## [Decision Letter · Decision Letter 1]

17 Jan 2022

Absence of association between host genetic mutations in the ORAI1 gene and COVID-19 fatality

PONE-D-21-24713R1

Dear Dr. Bailey,

We’re pleased to inform you that your manuscript has been judged scientifically suitable for publication and will be formally accepted for publication once it meets all outstanding technical requirements.

Kind regards,

Alessandro Marchioni

Academic Editor

PLOS ONE

Additional Editor Comments (optional):

Dear Professor Bailey,

I am pleased to inform that your paper "Absence of association between host genetic mutations in the ORAI1 gene and COVID-19 fatality" has been revised and considered suitable for publication in PLOS ONE.

CRAC channels have a critical role in the activation of T lymphocytes, and mutations in either Orai1 or STIM1 are characterized by severe immunodeficiency (SCID)-like disease and autoimmunity. Therefore, considering severe COVID-19 the results of a immune response dysregulation, the potential association of genetic polymorphisms of ORAI1 with the risk of Severe Acute Respiratory Syndrome related to SARS-CoV-2 (CARDS) is a relevant topic. Although the authors results suggest that in COVID-19 the host genetic polymorphisms of ORAI1 are unlikely to be implicated in symptoms severity among afflicted patients, we think that the informations provided in the paper could increases the knowledge about this complex disease.

Reviewers' comments:

Reviewer's Responses to Questions

**Comments to the Author**

1. If the authors have adequately addressed your comments raised in a previous round of review and you feel that this manuscript is now acceptable for publication, you may indicate that here to bypass the “Comments to the Author” section, enter your conflict of interest statement in the “Confidential to Editor” section, and submit your "Accept" recommendation.

Reviewer #1: All comments have been addressed

Reviewer #2: All comments have been addressed

2. Is the manuscript technically sound, and do the data support the conclusions?

Reviewer #1: (No Response)

Reviewer #2: Yes

3. Has the statistical analysis been performed appropriately and rigorously? 

Reviewer #1: Yes

Reviewer #2: Yes

4. Have the authors made all data underlying the findings in their manuscript fully available?

Reviewer #1: Yes

Reviewer #2: Yes

5. Is the manuscript presented in an intelligible fashion and written in standard English?

Reviewer #1: Yes

Reviewer #2: Yes

6. Review Comments to the Author

Reviewer #1: The authors answered my queries. Even in the presence of significant limitations of the study, I have no other comments to make and it can be published in its current form.

Reviewer #2: I would like to thank the Authors for their responses to my comments. I have no more comments to make.

7. PLOS authors have the option to publish the peer review history of their article (what does this mean?). If published, this will include your full peer review and any attached files.

Reviewer #1: No

Reviewer #2: **Yes: **Roberto Tonelli

---

## [Editor Report · Acceptance letter]

27 Jan 2022

PONE-D-21-24713R1 

Absence of association between host genetic mutations in the ORAI1 gene and COVID-19 fatality 

Dear Dr. Bailey:

I'm pleased to inform you that your manuscript has been deemed suitable for publication in PLOS ONE. Congratulations! Your manuscript is now with our production department. 

Kind regards, 

on behalf of

Dr. Alessandro Marchioni 

Academic Editor

PLOS ONE